# Profiling non-small cell lung cancer reveals that PD-L1 is associated with wild type EGFR and vascular invasion, and immunohistochemistry quantification of PD-L1 correlates weakly with RT-qPCR

**Akram Alwithenani[1,2], Drew Bethune[3], Mathieu Castonguay[1], Arik Drucker[4], Gordon Flowerdew[5], Marika Forsythe[1], Daniel French[3], John Fris[1], Wenda Greer[1], Harry Henteleff[3], Mary MacNeil[4], Paola Marignani[6], Wojciech Morzycki[4], Madelaine Plourde[3], Stephanie Snow[4], Paola Marcato[7]\*, Zhaolin Xu[1]\***

1 Department of Pathology, Faculty of Medicine, Dalhousie University Halifax, Nova Scotia, Canada, 2 Department of Laboratory Medicine, Faculty of Applied Medical Sciences, Umm Al-Qura University Makkah, Makkah, Saudi Arabia, 3 Department of Surgery, Faculty of Medicine, Dalhousie University Halifax, Nova Scotia, Canada, 4 Division Medical Oncology, Faculty of Medicine, Dalhousie University Halifax, Nova Scotia, Canada, 5 Department of Epidemiology, Faculty of Medicine, Dalhousie University Halifax, Nova Scotia, Canada, 6 Department of Biochemistry and Molecular Biology, Faculty of Medicine, Dalhousie University Halifax, Nova Scotia, Canada, 7 Departments of Pathology and Microbiology and Immunology, Dalhousie University, Halifax, Nova Scotia, Canada

\* zxu3@dal.ca (ZX); paola.marcato@dal.ca (PM)

**Data Availability Statement:** All relevant data are within the paper except for the S1 Table (S1 Table).

## Abstract

Most lung cancer patients are diagnosed at an advanced stage, limiting their treatment options with very low response rate. Lung cancer is the most common cause of cancer death worldwide. Therapies that target driver gene mutations (e.g. *EGFR*, *ALK*, *ROS1*) and checkpoint inhibitors such anti-PD-1 and PD-L1 immunotherapies are being used to treat lung cancer patients. Identification of correlations between driver mutations and PD-L1 expression will allow for the best management of patient treatment. 851 cases of non-small cell lung cancer cases were profiled for the presence of biomarkers *EGFR*, *KRAS*, *BRAF*, and *PIK3CA* mutations by SNaPshot/sizing genotyping. Immunohistochemistry was used to identify the protein expression of ALK and PD-L1. Total PD-L1 mRNA expression (from unsorted tumor samples) was quantified by RT-qPCR in a sub-group of the cohort to assess its correlation with PD-L1 protein level in tumor cells. Statistical analysis revealed correlations between the presence of the mutations, PD-L1 expression, and the pathological data. Specifically, increased PD-L1 expression was associated with wildtype EGFR and vascular invasion, and total PD-L1 mRNA levels correlated weakly with protein expression on tumor cells. These data provide insights into driver gene mutations and immune checkpoint status in relation to lung cancer subtypes and suggest that RT-qPCR is useful for assessing PD-L1 levels.

**Funding:** The study was partially supported by Pfizer Canada, Roche Canada, Boehringer Ingelheim Canada and Merck Canada. The funders had no role in study design, data collection and analysis, decision to publish, or preparation of the manuscript. The funding provided by the pharmaceutical companies were helping us for validation of the tests and not related to employment, consultancy, patents, products in development, marketed products, and any commercial purposes. This does not alter our adherence to PLOS ONE policies on sharing data and materials. Support was also provided by grant funding to Paola Marcato from the Canadian Institutes of Health Research (CIHR, MOP-130304 and PJT-162313). The funders had no role in study design, data collection and analysis, decision to publish, or preparation of the manuscript.

**Competing interests:** The study was partially supported by Roche Canada, Pfizer Canada and Boehringer Ingelheim Canada. The funders had no role in study design, data collection and analysis, decision to publish, or preparation of the manuscript. The funding provided by the pharmaceutical companies were helping us for validation of the tests and not related to employment, consultancy, patents, products in development, marketed products, and any commercial purposes. This does not alter our adherence to PLOS ONE policies on sharing data and materials.

# Introduction

Lung cancer is a leading cause of cancer death, killing more people than breast, prostate, and colorectal cancers combined [1]. Unfortunately, more than 50% of lung cancer patients die within one year of diagnosis [1]. Even in localized lung cancer, the five-year survival is only about 55%, suggesting that biomarker testing in the early stages of the disease has the potential to make a major improvement in the disease control and management. The advancements in molecular profiling in lung cancer have provided powerful tools for implementing new treatments, such as EGFR and ALK tyrosine kinase inhibitors. Patients with metastatic lung adenocarcinoma harbouring *EGFR* mutations or *ALK* rearrangements experience better quality of life, lower toxicity, and encouraging outcomes when they receive tyrosine kinase inhibitors [2]. However, patients treated with selective inhibitors experience tumor progression because of resistance-conferring secondary mutations [3]. In addition, a large proportion of lung cancers do not exhibit targetable driver mutations that have approved drugs by the Food and Drug Administration (FDA). Moreover, patients with *KRAS* mutations, the most common driver mutations in lung cancer, demonstrate low response to targeted therapies [4].

More recently, immunotherapy represents an exciting new approach in cancer treatment. Checkpoint inhibitors are currently used for lung cancer treatment. The main goal of immunotherapy is to boost the immune system by activating immune cells to recognize and kill tumor cells. T cells play a critical role in many immunotherapies, and their activation depends on three key signals. First is the interaction between the T cell receptor and the antigenic peptide-major histocompatibility complex. Second is antigen–independent costimulatory signals, which involve an activating signal like CD28, and an inhibitory signal, such as the PD-1 and cytotoxic T lymphocyte-associated antigen 4 receptor pathways. Third is cytokines, such as interferon gamma (IFN-γ), which is secreted by immune cells, and induces the expression of PD-L1. Many tumor cells that develop from organs such as lung, head and neck, colon, stomach, and skin, express PD-L1 [5]. Tumor cells evade immune surveillance via the interaction between PD-1 and PD-L1, which supress the activation of T cells. Generally, the interaction of PD-1 and PD-L1 plays a role in the inhibition of cell apoptosis, suppression of immune reaction to tumors, and tumor evasion of the immune system [6]. There are several reasons why inhibitors of PD-1/PD-L1 interaction are particularly promising anti-cancer immunotherapies. First, tumor-infiltrating lymphocytes and circulating tumor-specific T cells exhibit high expression of PD-1. Second, the correlation between the expression of PD-L1 and the prognosis of many cancers suggests that the expression of PD-L1 is a tumor mechanism for the evasion of immune surveillance [7]. There is controversy regarding the prognostic role of PD-L1 expression, as some authors have shown inferior outcomes when correlating with prognosis [8], and others observed improved outcomes [9]. Based on the existing evidence, PD-1 and PD-L1 inhibitors may play a role in breaking some of the multiple layers of immune inhibition and inducing an effective T cell response against tumors. Tumor cells have noticeably higher PD-L1 expression in comparison with adjacent lung parenchyma [10]. Additionally, PD-L1 expression is associated with poor prognosis and short overall survival [11]. Along with the new emerging checkpoint inhibitors in lung cancer, it is expected that increasing the overall survival rate in lung cancer will involve detecting particular targetable gene mutations and PD-1/PD-L1 expression. *EGFR* mutations are linked with good prognosis in lung cancer patients mainly attributed to the treatment of tyrosine kinase inhibitor (TKI) [12], but also seen in surgically resected non-small cell lung cancer (NSCLC) without receiving TKI [13]. Several reports have also shown association between the expression of PD-L1 and poor survival rate in lung cancer patients [14, 15]. It is not known whether RT-qPCR can be used as an alternative diagnostic method to detect PD-L1 expression by immunohistochemistry (IHC) in

lung cancer. Thus, we hypothesize that the membranous expression of PD-L1 on lung tumor cells using 22C3 antibody and/or the absence of *EGFR* mutations will be associated with unfavorable pathologic characteristics, and that *PD-L1* mRNA expression by RT-qPCR will correlate with PD-L1 protein expression using 22C3 anti-PD-L1 by IHC.

## Material and methods

### Study population

Samples from patients who underwent surgical resection for lung cancer from 2005 to 2017 at the Queen Elizabeth II (QEII) Health Sciences Centre in Halifax, Canada, were enrolled in the study. Nova Scotia Health Authority's Research Ethics Board approved the study and all patients provided written informed consent. A total of 851 cases with anonymized data formed the study cohort. Tumor samples included both fresh and formalin-fixed paraffin-embedded tissue (FFPE). A 4μm-thick section from each FFPE tissue block was mounted on a glass slide and stained with hematoxylin and eosin (H&E). An appropriate tumor tissue block was chosen for further studies. All cases had undergone molecular profiling using two set tests. First, a multiplex polymerase chain reaction-based assay (SNaPshot platform) [16] to detect a panel of point mutations in commonly mutated genes, including *EGFR*, *KRAS*, *BRAF*, and *PIK3CA*. According to the manufacturer's instructions (ABI PRISM SNaPshot Multiplex Kit cat#4323151) and products were resolved on an ABI 3130XL capillary sequencer (Applied Biosystems). Second, quadruplex fragment analysis genotyping to detect deletion and insertion mutations at exons 19 and 20 in the *EGFR* gene using differentially labelled fluorescent PCR primers specific for regions that flank the deletion/insertion sites to generate amplicons that are sized and detected using a capillary sequencer. Demographic information, clinicopathological data (including age, sex, cancer subtype, vascular invasion, lymphatic invasion, lymph node metastasis, staging, and smoking history), and mutational status were retrieved from laboratory files and medical records.

In a subset of the cohort, 232 FFPE lung tumor samples, were used to quantify PD-L1 protein utilizing IHC and 49 fresh tumor samples were used to quantify certain immune-related genes including *PD-L1* mRNA utilizing real time quantitative polymerase chain reaction (RT-qPCR).

### Immunohistochemistry

For PD-L1 expression analysis, IHC using an automated stainer (Link 48, Dako) was performed on 4μm sections cut from archival FFPE tumor samples from 232 patients diagnosed with non-small cell lung cancer that were retrospectively selected from the QEII Health Sciences Centre. PD-L1 IHC using the PD-L1 22C3 pharmDx kit on the Dako platform (Product number: SK006) was performed according to manufacturer recommendations [17]. The positive and negative controls were from known PD-L1 IHC positive and negative cases confirmed by IHC testing. The pharmDx kit (Dako) is designed to perform the staining using a linker and a chromogen enhancement reagent. Pre-treatment of the slides including deparaffinization and rehydration was performed using PT Link machine. Next, the specimens were incubated with monoclonal mouse IgG antibody to PD-L1, followed by incubation with a mouse linker and with a ready-to-use Visualization Reagent consisting of Goat secondary antibodies against mouse immunoglobulin and horseradish peroxidase coupled to a dextran polymer backbone. Then, chromogen and chromogen enhancement reagents were added, resulting in a brown color at the site of the antigen-antibody interaction. All slides were cover slipped and visualized with a light microscope.

### Interpretation of PD-L1 expression by immunohistochemistry

Each PD-L1 stained slide had a paired H&E slide from the same block in order to identify the tumor cells precisely. PD-L1 protein expression is determined by a Tumor Proportion Score (TPS), which is the percentage of viable tumor cells showing partial or complete membrane staining. We used 1% and 50% cut-offs for PD-L1 expression to align with current clinical practice and clinical significance [18, 19]. All IHC numerations and analyses were conducted by lung pathologists (Z.X. and M.C).

### Quantitative PCR

Total RNA from fresh tumor samples was extracted using TRIzol (Invitrogen) and the Pure-link RNA kit (Invitrogen) with DNase treatment. Equal amounts of RNA were reverse-transcribed using iScript (BioRad) and quantitative real-time PCR was performed using gene-specific primers. Standard curves for each primer set were generated, and primer efficiencies were incorporated into the CFX Manager software (Bio-Rad). Relative levels of mRNA were calculated utilizing internal reference genes TATA-Box Binding Protein (*TBP*) and Ribosomal Protein L13a (*RPL13A*). Relative mRNA expression was log-2 transformed prior to plotting and statistical analysis. The primer sequences are listed in S1 Table.

### Statistical analysis

Two software programs were used to do the analysis, Statistical Analysis System (SAS) 9.3 (version 14.0; StataCorp, College Station, TX) and GraphPad Prism (GraphPad, San Diego, USA). SAS 9.3 was used because it is the most appropriate software to analyze clinical data for large cohorts and GraphPad Prism was used for two variable comparison and to generate graphs. The association between the gene mutations, PD-L1 expression, and clinicopathological features was evaluated. Statistical analysis was performed using SAS 9.3 software. Categorical variables were compared using the Pearson's goodness-of-fit test or Fisher's exact tests, as appropriate, and continuous variables were analyzed using a Wilcoxon rank-sum test (Mann-Whitney U test). Statistical comparisons were made by a two-tailed Student's t-test, Spearman correlation using GraphPad Prism software. All hypothesis tests were two-sided, and a p value less than 0.05 was considered statistically significant.

## Results

### Patient characteristics

Our group has recently profiled a large Nova Scotian lung cancer patient cohort for major driver mutations in lung cancer (*EGFR*, *ALK*, *KRAS*, *BRAF*, and *PIK3CA*) [20]. Here we assess the relationship between clinicopathological data, driver mutations, and PD-L1 expression in the expanded patient cohort. Additionally, we assessed the possibility of using RT-qPCR as a method to detect PD-L1 expression in patient samples by comparing the levels of PD-L1 detected with the IHC data.

In a total of 851 eligible patients with non-small cell lung cancer, the vast majority had adenocarcinoma histology (65%). The rest were divided between squamous cell carcinoma (24%), large cell carcinoma (6%), and rare subtypes (5%). Most of the patients were stage I (56%), followed by stage II (26%), stage III (16.3%), and stage IV (1.4%). Men and women represented equal proportions (49% and 51%, respectively). The mean age at diagnosis was 66 years (range, 34–90). The frequency of specific gene mutations was investigated; of 851 lung cancer patients, 552 were lung adenocarcinoma, in which specific gene mutations were identified in 270 patients. These included 199 *KRAS* mutations, 55 *EGFR* mutations, 6 *PIK3CA* mutations, 9

*BRAF* mutations, and one *ALK* rearrangement. The details of molecular alterations including all lung cancer subtypes are described in Table 1. Two patients exhibited two mutations (*EGFR* & *PIK3CA* and *KRAS* & *PIK3CA*).

## Correlation between clinicopathologic features and classical driver mutations

Clinicopathologic characteristics were correlated with classical driver mutations such as *KRAS* and *EGFR* mutations. In Table 2 we show a summary of all significant associations between variables and gene mutations in the lung cancer patient cohort. *EGFR* mutations were significantly associated with female versus male patients ($p<0.001$). *KRAS* mutations were more prevalent in the younger group, ranging from 34 to 59 years ($p = 0.03$, Table 2). In addition, never smokers with non-small cell lung cancer were significantly associated with *EGFR* mutations ($p<0.001$). These clinical variables are summarized in Table 3. Significant associations between mutations and lymph-vascular invasion and tumor grade could indicate a poor or good prognostic status. The absence of vascular invasion was associated with *EGFR* mutations ($p<0.01$). However, pleural or lymphatic invasion and with lymph nodes metastasis has shown negative correlation with EGFR mutations. In addition, no positive correlation was reported between these driver mutations and tumor grade. All pathological features are shown in Table 4. Table 2 includes only the significant associations between the variables and gene mutations. Well-differentiated histology was significantly associated with *EGFR* mutations, but not so for *KRAS* mutations ($p<0.001$). Poorly differentiated histology was associated with the absence *EGFR* and *KRAS* mutations ($p<0.001$). Patients with lung adenocarcinoma were significantly associated with *KRAS* and *EGFR* mutations ($p<0.001$), but other subtypes such as squamous cell and large cell carcinomas, were associated with the absence of *KRAS* and *EGFR* mutations ($p<0.001$) (Tables 2 and 5).

## PD-L1 expression on tumor cells is associated with more invasive disease

To determine if PD-L1 protein expression on tumor cells correlates with clinicopathologic characteristics, we performed IHC on a portion of lung cancer patient tumor samples. Of the 232 lung cancer cases (100 males and 132 females with the median age of 67), 114 (49%) cases

**Table 1. Details of molecular alterations in lung adenocarcinoma cohort.**

| Mutation | N, (%) |
|---|---|
| *KRAS* mutations | 199 (36) |
| G12X | |
| *EGFR* mutations | 55 (10) |
| L858R | 24 |
| Exon 19 deletions | 28 |
| Exon 20 insertions | 3 |
| *BRAF* mutations | |
| V600E | 9 (2) |
| *PIK3CA* mutations | 6 (1) |
| E545K | 4 |
| E542K | 2 |
| *ALK* rearrangements | 1 (0.2) |
| Total | 270 (49) |

(%) percentage of total lung adenocarcinoma cases.

**Table 2. A summary of all significant association between variables and gene mutations in lung cancer patients cohort.**

| | | *EGFR* mutations | | | *KRAS* mutations | | |
|---|---|---|---|---|---|---|---|
| | N | Observed | Expected# | p | Observed | Expected | p |
| Age < 59 | 179 | 16 | 12.4 | | 55 | 44.6 | * |
| Male | 416 | 15 | 28.8 | *** | 92 | 103.6 | |
| Female | 435 | 44 | 30.2 | *** | 120 | 108.4 | |
| Vascular invasion | 362 | 14 | 25.1 | ** | 101 | 90.2 | |
| No vascular invasion | 489 | 45 | 33.9 | ** | 111 | 121.8 | |
| Smoked | 668 | 27 | 41.2 | *** | 175 | 170.3 | |
| Never smoked | 46 | 17 | 2.87 | *** | 7 | 11.7 | |
| Adenocarcinoma | 552 | 56 | 38.5 | *** | 199 | 137.0 | *** |
| Squamous cell | 205 | 1 | 14.3 | *** | 4 | 50.9 | *** |
| Large cell carcinoma | 51 | 0 | 3.6 | | 3 | 12.7 | ** |
| Well differentiated | 85 | 15 | 5.9 | *** | 21 | 16.3 | |
| Moderately differentiated | 320 | 34 | 22.3 | ** | 81 | 61.3 | *** |
| Poorly differentiated | 441 | 10 | 30.8 | *** | 60 | 84.4 | *** |

* $p < 0.05$ (two-tail);

** $p < 0.01$ (two-tail);

*** $p < 0.001$ (two-tail)—P-values obtained from Pearson's goodness-of-fit test.

# In any given table, expected values are calculated by multiplying the total number of the raw with the total number of the column divided by overall total number of the table.

demonstrated PD-L1 membranous staining on tumor cells using 1% as a cut-off (almost half of patients) and 118 (51%) showed PD-L1 expression < 1%. Therefore, 1% cut-off represents the median for PD-L1 membrane staining in the cohort. Of 232 patients, pathologic staging was available for 163 and smoking data were available for 162. One hundred and fifty-four

**Table 3. Clinical characteristics of patients with NSCLC (N = 851) and their relationship with the most common gene mutations.**

| Parameter | Gene mutation | | | | | | P value |
|---|---|---|---|---|---|---|---|
| | *ALK* | *EGFR* | *KRAS* | *BRAF* | *PIK3CA* | None identified | |
| **Sex, N (%)** | | | | | | | < 0.001 |
| Female | 1 (0.2) | 44 (10.1) | 120 (27.6) | 3 (0.6) | 7 (1.6) | 260 (59.8) | |
| Male | 0 (0) | 15 (3.6) | 92 (22.1) | 6 (1.4) | 5 (1.2) | 298 (71.6) | |
| **Age** | | | | | | | 0.122 |
| <50 | 0 (0) | 16 (8.9) | 55 (30.7) | 0 (0) | 2 (1.1) | 106 (59.2) | |
| 60–74 | 1 (0.2) | 29 (5.9) | 117 (23.7) | 4 (0.8) | 8 (1.6) | 335 (67.8) | |
| >75 | 0 (0) | 14 (7.9) | 40 (22.5) | 5 (2.8) | 2 (1.1) | 117 (65.7) | |
| **Smoking** | | | | | | | <0.0001 |
| Never Smoked | 0 (0) | 17 (42.5) | 7 (17.5) | 1 (2.5) | 0 (0) | 21 (52.5) | |
| Smoked | 1 (0.2) | 27 (4.3) | 175 (27) | 7 (1.1) | 9 (1.4) | 449 (70.8) | |
| **Stage** | | | | | | | 0.146 |
| I | 1 (0.2) | 29 (7.2) | 108 (26.8) | 7 (1.7) | 3 (0.7) | 255 (63.3) | |
| II | 0 (0) | 6 (3.2) | 37 (19.6) | 1 (0.5) | 4 (2.1) | 141 (74.6)) | |
| III | 0 (0) | 8 (6.8) | 32 (27.4) | 0 (0) | 2 (1.7) | 75 (64.1) | |
| IV | 0 (0) | 1 (11.1) | 2 (22.2) | 0 (0) | 0 (0) | 6 (66.7) | |

P-values obtained from Pearson's goodness-of-fit test after pooling ALK, BRAF, P1K3CA and unknown mutations into a single category (Other) so that the expected count in each cell is at least 5.

**Table 4. Poor prognosis factors of patients with NSCLC (N = 851) and their relationship with the most common gene mutations.**

| Parameter | Gene Mutation | | | | | | |
|---|---|---|---|---|---|---|---|
| | ALK | EGFR | KRAS | BRAF | PIK3CA | None identified | P value |
| **Pleural invasion, N (%)** | | | | | | | 0.285 |
| No | 1 (0.2) | 46 (7) | 171 (26.1) | 7 (1.1) | 6 (0.9) | 423 (64.7) | |
| Yes | 0 (0) | 13 (6.6) | 41 (20.8) | 2 (1) | 6 (3) | 135 (68.5) | |
| **Vascular invasion** | | | | | | | **0.004** |
| No | 0 (0) | 45 (9.2) | 111 (22.7) | 6 (1.2) | 3 (0.6) | 324 (66.3) | |
| Yes | 1 (0.3) | 14 (3.9) | 101 (27.9) | 3 (0.8) | 9 (2.5) | 234 (64.6) | |
| **Lymphatic invasion** | | | | | | | 0.199 |
| No | 1 (0.2) | 45 (8.1) | 138 (24.7) | 6 (1.1) | 6 (1.1) | 362 (64.9) | |
| Yes | 0 (0) | 14 (4.8) | 74 (25.3) | 3 (1) | 6 (3) | 196 (66.9) | |
| **Lymph nodes** | | | | | | | 0.706 |
| N0 | 1 (0.2) | 41 (7) | 149 (25.6) | 8 (1.4) | 8 (1.4) | 375 (64.4) | |
| N1 | 0 (0) | 10 (5.8) | 37 (21.6) | 1 (0.6) | 4 (2.3) | 119 (69.6) | |
| N2 | 0 (0) | 8 (8.2) | 26 (26.5) | 0 (0) | 0 (0) | 64 (65.3) | |

P-values obtained from Pearson's goodness-of-fit test after pooling ALK, BRAF, P1K3CA and unknown mutations into a single category (Other) so that the expected count in each cell is at least 5.

(95%) patients were smokers. Stage I disease occurred in 92 (56.4%) while stage II and III occurred in 40 (24.5%) and 28 (17.2%) respectively. Only 3 (1.8%) patients were at stage IV. Some of the clinicopathologic features of the cohort were correlated with PD-L1 expression using 1% cut-off. There was no significant association between PD-L1 expression and age, sex, pathological stage and smoking status. Greater than 1% PD-L1 membranous expression on tumor cells was significantly associated with vascular invasion ($p = 0.035$), but not pleural invasion, lymphatic invasion, or lymph nodes metastasis. PD-L1 expression was shown negative correlation with lymph nodes involvements and tumor size as well (Table 6).

**EGFR mutations are associated with the absence of PD-L1.** We also assessed the association between PD-L1 membranous staining on tumor cells using 1% cut-off and the presence of the *EGFR* and *KRAS* mutations. Molecular alterations were identified in 114 (49%) of the PD-L1 stained sub-cohort, including 78 *KRAS* mutations, 23 *EGFR* mutations, 5 *BRAF* mutations, and 8 *PIK3CA* mutations. PD-L1 expression was present in 44 (56%) *KRAS* mutants, but only in 6 (26%) *EGFR* mutants. Therefore, *EGFR* mutations were significantly associated with the absence of PD-L1 expression ($p = 0.02$, Fig 1). However, there was no significant association between *KRAS* mutations and the expression of PD-L1 ($p = 0.10$, Fig 1).

## PD-L1 expression by IHC correlates with *PD-L1* mRNA expression by RT-qPCR

Here, we aimed to investigate the feasibility of using RT-qPCR as a diagnostic tool in the quantification of PD-L1 and the correlation between immune-relating genes (*CD3*, *CD8*, and *CD45*) and PD-L1. The first objective is to investigate the expression of PD-L1 and other immune related genes by RT-qPCR in fresh lung samples obtained from lung cancer patients at the QEII Health Sciences Centre, Halifax, Canada. The second objective is to see if the levels of PD-L1 and immune related markers detected by RT-qPCR correlate with PD-L1 by IHC. It is notable that for the RT-qPCR, we are assessing total PD-L1 and not just PD-L1 that is specific to tumor cells as was quantified by IHC. This is because if RT-qPCR is to be used as a

**Table 5. Pathological characteristics of patients with NSCLC (N = 851) and their relationship with the most common gene mutations.**

| Parameter | Gene Mutation | | | | | | P value |
|---|---|---|---|---|---|---|---|
| | ALK | EGFR | KRAS | BRAF | PIK3CA | None identified | |
| **Location, N (%)** | | | | | | | 0.985 |
| RUL | 1(0.3) | 21(6.9) | 75 (24.6) | 3 (1) | 3 (1) | 202 (66.2) | |
| RU+ML | 0 (0) | 0 (0) | 1 (20) | 0 (0) | 0 (0) | 4 (80) | |
| RML | 0 (0) | 3 (7.9) | 11 (28.9) | 2 (5.3) | 1 (2.6) | 21 (55.3) | |
| RM+LL | 0 (0) | 0 (0) | 1 (33.3) | 0 (0) | 0 (0) | 2 (66.7) | |
| RLL | 0 (0) | 11 (8.3) | 35 (26.3) | 1 (0.8) | 2 (1.5) | 84 (63.2) | |
| RUL+RML+RLL | 0 (0) | 0 (0) | 3 (27.3) | 0 (0) | 0 (0) | 8 (72.7) | |
| LUL | 0 (0) | 15 (6.7) | 58 (25.8) | 3 (1.3) | 4 (1.8) | 145 (64.4) | |
| LLL | 0 (0) | 9 (7.8) | 25 (21.6) | 0 (0) | 2 (1.7) | 80 (69) | |
| LLL+LUL | 0 (0) | 0 (0) | 1 (9.1) | 0 (0) | 0 (0) | 10 (90.9) | |
| **Cell type** | | | | | | | <0.0001 |
| AD | 0 (0) | 56 (10.1) | 199 (36.1) | 9 (1.6) | 6 (1.1) | 282 (51.1) | |
| ADSQ | 0 (0) | 0 (0) | 1 (14.3) | 0 (0) | 0 (0) | 6 (85.7) | |
| SQ | 0 (0) | 1 (0.5) | 4 (2) | 0 (0) | 5 (2.4) | 195 (95.1) | |
| LCC | 0 (0) | 0 (0) | 3 (5.9) | 0 (0) | 1 (2) | 47 (92.2) | |
| PLC | 0 (0) | 0 (0) | 2 (15.4) | 0 (0) | 0 (0) | 11 (84.6) | |
| Carcinoid | 0 (0) | 0 (0) | 0 (0) | 0 (0) | 0 (0) | 14 (100) | |
| AD in situ | 0 (0) | 2 (50) | 1 (25) | 0 (0) | 0 (0) | 1 (25) | |
| **Differentiation** | | | | | | | <0.0001 |
| W | 0 (0) | 15 (17.6) | 21 (24.7) | 0 (0) | 0 (0) | 49 (57.6) | |
| M | 1 (0.3) | 34 (10.6) | 81 (25.3) | 2 (0.6) | 6 (1.9) | 196 (61.3) | |
| P | 0 (0) | 10 (2.3) | 60 (13.6) | 3 (0.7) | 9 (2) | 359 (81.4) | |

RUL: Right upper lobe; RU+ML: Right upper and Middle lobe; RML: Right middle lobe; RM+LL: Right middle and lower lobe; RLL: Right lower lobe; LUL: Left upper lobe; LLL: Left lower lobe; AD: Adenocarcinoma; ADSQ: Adenosquemous carcinoma; SQ: Squamous carcinoma; LCC: Large cell carcinoma; PLC: pleomorphic carcinoma; W: Well differentiated; M: Moderately differentiated; P: Poorly differentiated.

P-values obtained from Pearson's goodness-of-fit test after pooling ALK, BRAF, P1K3CA and unknown mutations into a single category (Other) so that the expected count in each cell is at least 5.

clinical method for quantification of PD-L1, it would be assessed from total/unsorted tumor samples. Forty-nine lung tumor samples were quantified for *PD-L1* mRNA transcriptional levels and three other immune-related genes (*CD3*, *CD8*, and *CD45*) utilizing RT-qPCR. The forty-nine tumor samples were previously quantified for PD-L1 membranous protein utilizing IHC. Comparing PD-L1 protein expression and *PD-L1* mRNA level revealed a good correlation (Spearman, $r = 0.29$, $p = 0.03$). In addition, correlation between PD-L1 on tumor cells including immune cells and RT-qPCR of *PD-L1* mRNA level was higher (Spearman, $r = 0.31$, $p = 0.02$, Fig 2). This suggests the possibility of using RT-qPCR as an alternative method for detection of PD-L1 in non-small cell lung cancer cases; however, how RT-qPCR detected-PD-L1 correlates with response to therapy will need to be determined.

Looking at the correlation with other markers (CD45, CD3, CD8) and levels of PD-L1 by IHC could help identify a significant marker that has a role in predicting response to checkpoint inhibitors along with PD-L1. CD45, which is a general biomarker for leukocytes, including T and B cells, showed no significant correlation with PD-L1 detected by IHC for 1% and 50% cut-offs ($p = 0.49$; $p = 0.12$). Likewise, CD3, a marker for T cells including T helper cells and T cytotoxic cells, demonstrated no significant correlation with PD-L1 detected by IHC for

**Table 6. Clinicopathological characteristics and molecular alterations of lung adenocarcinoma patients stratified by PD-L1 expression on tumor cells.**

| Variable | PD-L1 expression (≥1% vs. <1%) | | |
|---|---|---|---|
| | PD-L1+ N (%) | PD-L1- N (%) | *p* |
| All patients | 114 | 118 | |
| Sex# | | | 0.069 |
| Female | 58 (51) | 74 (63) | |
| Male | 56 (49) | 44 (37) | |
| Age | | | 0.902 |
| < 60 | 23 (20) | 25 (21) | |
| 60–74 | 68 (60) | 67 (57) | |
| >75 | 23 (20) | 26 (22) | |
| Smoking # | | | 0.065 |
| Never Smoked | 1 | 7 | |
| | 77 | 77 | |
| Tumor size in cm (IQR)[1] | 2.4 | 2 | 0.851 |
| T status (pT)[2] | | | 0.255 |
| T1 | 41 (36) | 49 (41) | |
| T2 | 50 (44) | 53 (45) | |
| T3 | 18 (16) | 9 (8) | |
| T4 | 5 (4) | 7 (6) | |
| N status (pN)[3] | | | 0.856 |
| N0 | 79 (71) | 78 (68.4) | |
| N1 | 19 (17) | 23 (20.2) | |
| N2 | 13 (12) | 13 (11.4) | |
| Pathologic Stage | | | 0.830 |
| I | 44 (56) | 48 (56) | |
| II | 21 (27) | 19 (22) | |
| III | 12 (15) | 16 (19) | |
| IV | 1 (1) | 2 (2) | |
| Pleural invasion[4] # | | | 0.060 |
| 0 | 72 (37) | 88 (75) | |
| 1 | 42 (63) | 30 (25) | |
| Lymphatic invasion # | | | 0.057 |
| 0 | 68 (61) | 85 (72) | |
| 1 | 45 (39) | 33 (28) | |
| Vascular invasion # | | | 0.035 |
| 0 | 47 (41) | 65 (55) | |
| 1 | 67 (59) | 53 (45) | |

[1] interquartile range.

[2] $T_1$ = tumor 3 cm or less; $T_2$ = tumor more than 3 cm but ≤ 7 cm; $T_3$ = tumor more than 7 cm; $T_4$ = tumor of any size that invades any of the following: mediastinum, heart, great vessels, and trachea.

[3] N0 = no tumor cells in lymph nodes. N1 = tumor cells present in ipsilateral peribronchial, hilar and intrapulmonary nodes, N2 = tumor cells present in ipsilateral mediastinal and subcarinal nodes.

[4] 0 = absent; 1 = present. P values were obtained from Pearson's goodness-of-fit test and Fisher's exact test (#).

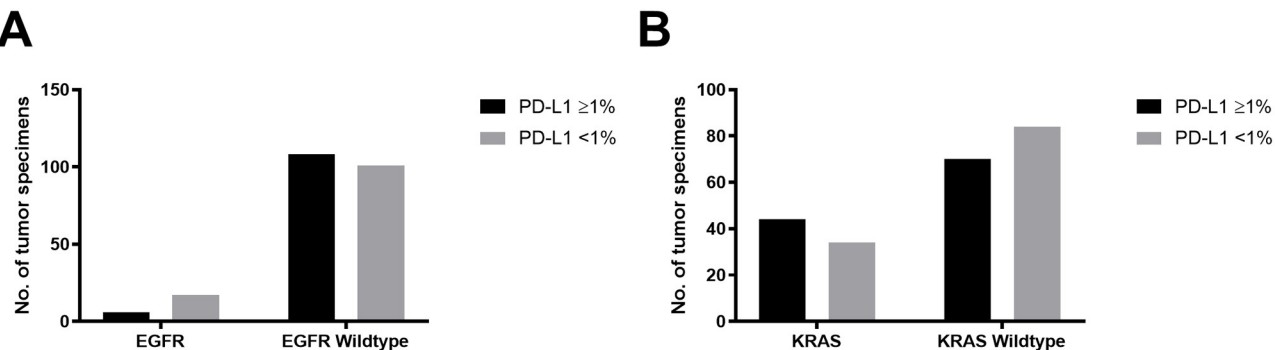

**Fig 1. *EGFR* but not *KRAS* was negatively correlated with PD-L1 membranous protein expression in the lung cancer patient's cohort.** A total of 232 lung tumors were evaluated for PD-L1 expression on tumor cells. All patients were screened previously for molecular alterations. EGFR positive patients were shown to negatively correlated to PD-L1 ($p = 0.02$; Fisher exact test) 26% of EGFR+ patients had PD-L1 expression versus 74% had PD-L1 expression in the same population.

1% and 50% cut-offs ($p = 0.47$; $p = 0.25$). However, CD8, a biomarker for T cytotoxic cells, correlated with PD-L1 by IHC for 50% cut-off ($p = 0.04$) but not for 1% cut-off ($p = 0.57$, Fig 3).

## Discussion

Our study demonstrates data on the frequency of *KRAS* and *EGFR* mutations in a large cohort of patients diagnosed with non-small cell lung cancer that underwent surgical resection treatment over a defined period in Halifax, Canada. The frequency of *EGFR* mutations in our study was reported at 7%. This rate was different than other studies reported in the literature. For instance, an *EGFR* mutational rate of 16.6% was reported in a cohort consisting of 2105 lung cancer patients from 126 hospitals in Spain, where an extensive study analyzed the frequency of *EGFR* mutations during the period of 2005–2008 [21]. One possible explanation for the higher rate of *EGFR* mutation could be differences in histological subgroups proportions, as the study demonstrated up to 78% of adenocarcinoma subgroup in comparison with our cohort that reported 65%. Considering that *EGFR* mutations are more common in adenocarcinomas and our cohort reported more than 90% of *EGFR* mutations in adenocarcinoma. Furthermore, another possible explanation is that many of the lung cancer patients in the Spanish cohort were diagnosed at later stage and biopsy specimens were used for the molecular alterations analysis, while lung cancer patients enrolled in our study were at relatively early stages and only surgical resections were used for assessment. With respect to *KRAS* mutational rate, our cohort reported 25%, which appears to be comparable with the Sequist et al. cohort study published on lung cancer patients and with other studies as well [22, 23]. Therefore, our *KRAS* mutations frequency is consistent with published reports.

The frequency of some of the molecular alterations in our cohort is relatively low. For instance, we have only one patient tumour out of 851 lung cancer patients that exhibited *ALK* rearrangement (0.12%), while other studies report a frequency of 3 to 6% [24]. In addition, our cohort has only 1.1% *BRAF* mutations which is considered to be a low percentage in comparison with other studies [25, 26]. Those low frequencies of *ALK* rearrangement and *BRAF* mutations could be attributed to the type of samples in our study, as we only have surgical resection samples and most of the patients were at early stages of lung cancer. Thus, the frequency of these mutations could increase if we include lung cancer patients from all stages, not only patients who treated with surgery at early stages. Additionally, regarding *BRAF* mutations, in

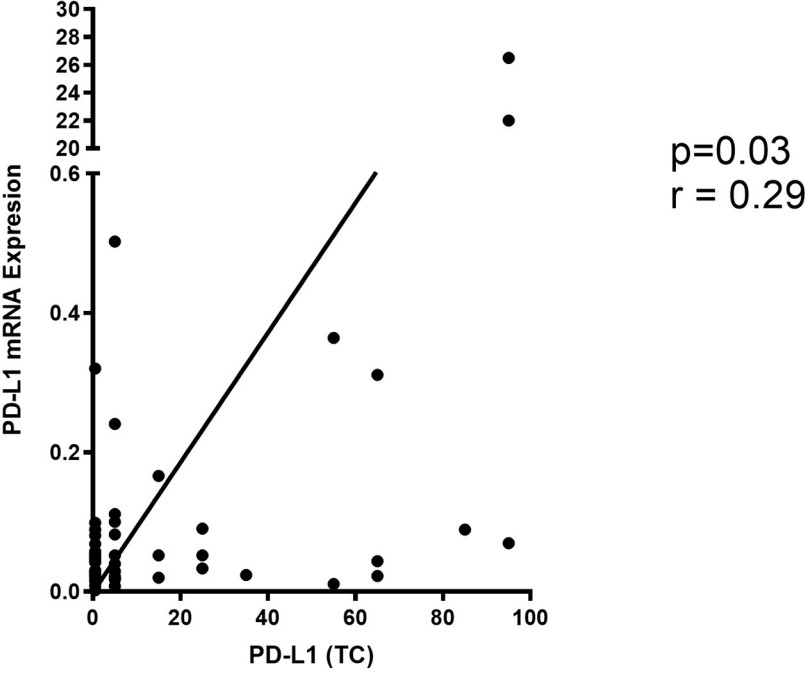

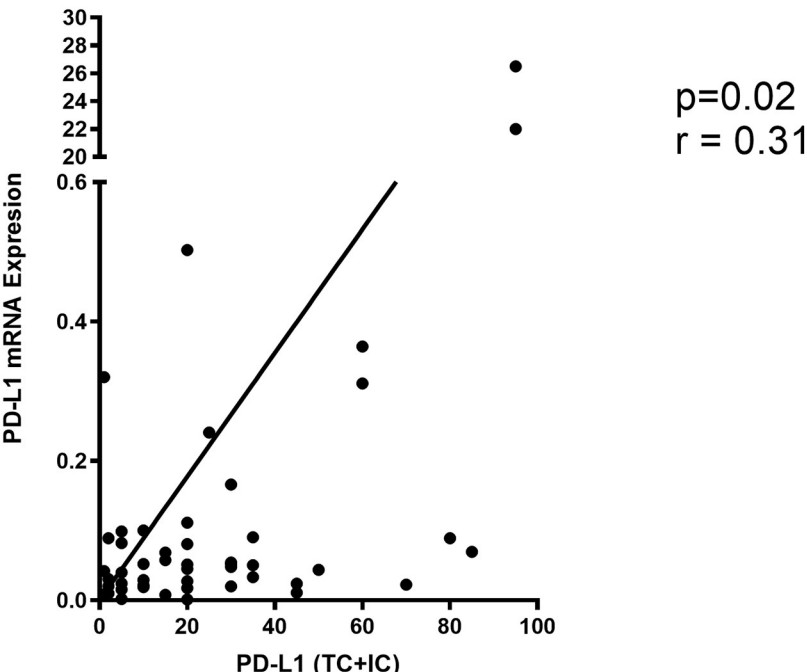

**Fig 2. PD-L1 expression by IHC correlates with *PD-L1* mRNA expression by qPCR.** A total of 49 fresh lung tumors were evaluated for PD-L1 expression by IHC and quantified for *PD-L1* mRNA by RT-qPCR. PD-L1 expression was evaluated on tumor cells only (TC), and on both tumor and immune cells (TC+IC). (A) PD-L1 expression on tumor cells (IHC) is significantly correlated with *PD-L1* mRNA expression (qPCR). (B) Also, PD-L1 expression tumor and immune cells (IHC) is significantly correlated with *PD-L1* mRNA expression (RT-qPCR).

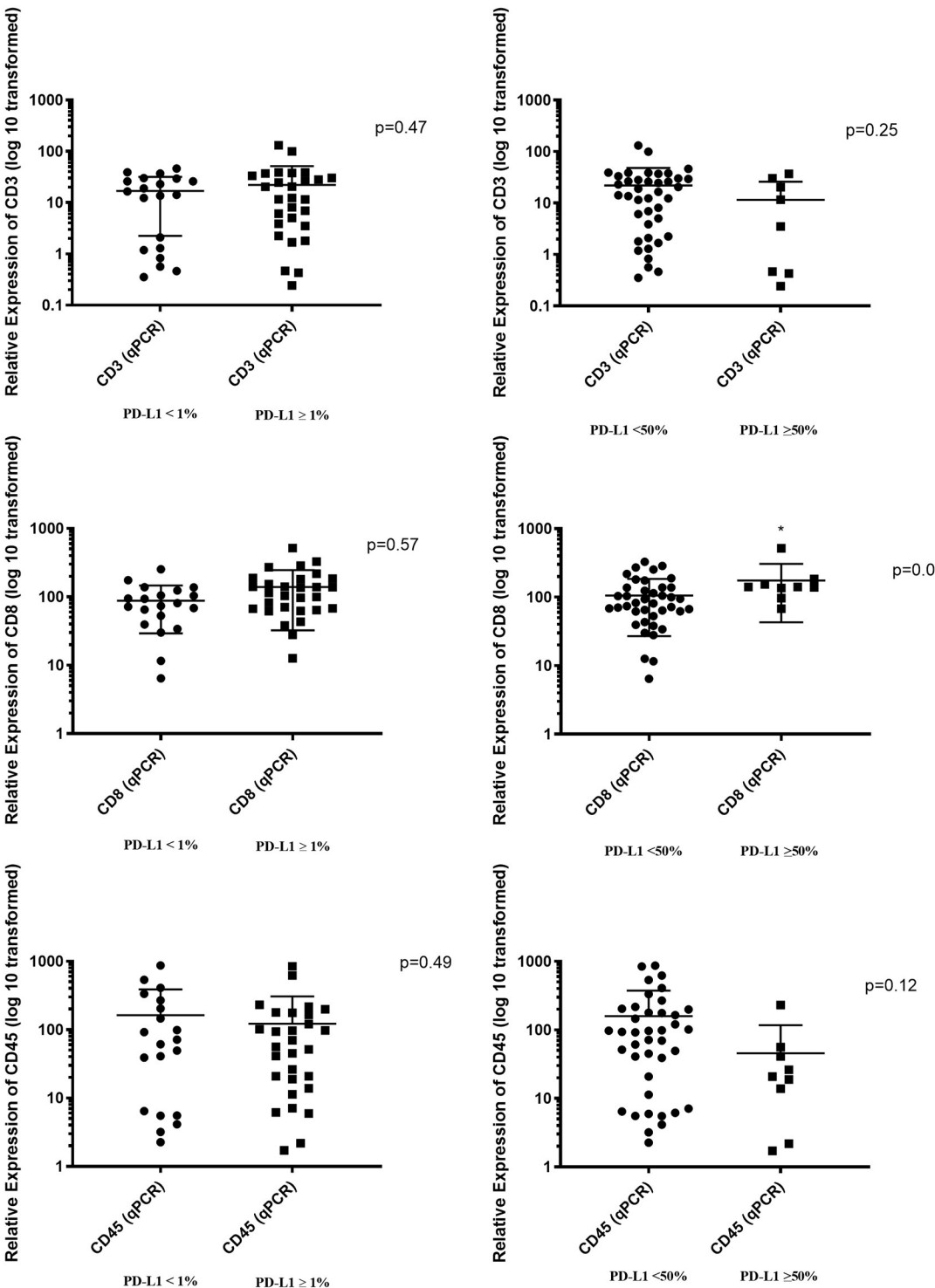

**Fig 3.** ***CD8* expression by qPCR correlates with PD-L1 expression by IHC for 50% cut-off.** A total of 49 fresh lung tumors were evaluated for PD-L1 expression by IHC and quantified for *CD8*, *CD3* and *CD45* mRNA by RT-qPCR. (A) *CD45* (RT-qPCR) did not correlate with PD-L1 (IHC, 1% and 50% cut-off). (B) Also, *CD3* marker (qPCR) was not significantly correlated with PD-L1 (IHC, 1% and 50% cut-off). (C) *CD8* marker (RT-qPCR) was significantly correlated with PD-L1 (IHC) for 50% cut-off but not for 1% cut-off.

our cohort, we only screened for V600E mutation which accounts for about 50% of all mutations in *BRAF* gene [27].

In this study of surgically resected lung cancer cases, we showed that membranous PD-L1 on tumor cells was associated with vascular invasion and marginally associated with pleural and lymphatic invasion. The presence of tumor cells in pleura, blood vessels, or lymphatics is an indication of poor prognosis and may contribute to metastases. There have also been several reports that indicate the association between PD-L1 and poor overall survival in non-small cell lung cancer [14, 28, 29].

There are two major mechanisms of PD-L1 over-expression in tumor cells: a) innate immune resistance and b) adaptive immune resistance [30]. In innate immune resistance, PD-L1 expression can be upregulated on tumor cells by constitutive oncogenic signaling independent of inflammatory signals in the tumor microenvironment. Non-small cell lung cancer models that harbour *EGFR* mutations and *ALK* rearrangements have demonstrated induction of PD-L1 expression and reduction of PD-L1 when treated with targeted therapies such as EGFR and ALK inhibitors [31, 32]. Furthermore, several clinical studies reported the association between PD-L1 expression and *EGFR* mutations and *ALK* fusions [32–34]. However, Zhang and colleagues showed that there was not an association between *EGFR* mutations and *ALK* rearrangements and PD-L1 expression [28].

Some studies have reported lack of associations between PD-L1 expression and *EGFR* status in lung cancer patients [35]. In this study we found that PD-L1 expression in at least some lung cancer cases was associated with wild-type *EGFR*. Zhang M et al., performed a meta-analysis of over 11,000 lung cancer patients from 47 studies and concluded the unfavourable prognostic values of PD-L1 as well as the correlation between PD-L1 expression and *EGFR* wild-type status [36]. This observation is consistent with the previously mentioned adaptive immune resistance, where the induction of PD-L1 expression is influenced by cytokines such as IFN-γ that is secreted from lymphocytes within the tumor microenvironment [37]. It is worth noting that due to low number of patients harbouring *BRAF*, *ALK* and *PIK3CA* mutations, we could not analyze the association between PD-L1 expression and these mutations.

In the clinical setting, the current method used for the detection of PD-L1 is IHC. In fact, among several agents targeting the PD-1/PD-L1 pathway, Pembrolizumab is the only drug approved by Health Canada and the FDA to treat metastatic non-small cell lung cancer in a first line setting. This is in association with a companion diagnostic test by IHC (anti-PD-L1 22C3 pharmDx) using the Dako Autostainer (Dako, Carpinteria, CA). It is worth noting, there are several other antibodies that have been validated for use in PD-L1 detection, such as Ventana SP142, Ventana SP263 [38]. In our study, we aimed to evaluate the possibility and the feasibility of using RT-qPCR to determine *PD-L1* mRNA expression in comparison with the IHC FDA-approved diagnostic test, as RT-qPCR could offer an efficient cost-effective method that provides information on the level of expression of PD-L1. Our results show that PD-L1 expression in tumor samples correlates significantly between RT-qPCR and IHC quantification methods. Significant correlation between PD-L1 protein expression by IHC and mRNA by RT-qPCR in bladder urothelial carcinoma has previously reported, using anti-PD-L1 E1L3N antibody [39], indicating a strong biological link between mRNA and protein expression regardless of the variation in the methodologies. Our study demonstrates a significant correlation between mRNA expression of *PD-L1* utilizing RT-qPCR and protein expression utilizing anti-PD-L1 22C3 pharmDx (IHC) and highlights the feasibility of using RT-qPCR as a potential method to detect PD-L1. RT-qPCR as a method of detection is faster than IHC and does not require a board-certified pathologist to diagnose each sample. The detection of PD-L1 expression on both tumor and immune cells has revealed clinical significance [40] and thus RT-qPCR which provides a total level of PD-L1 transcript, not specified to the tumor or

immune cell population has some clinical relevance. We currently lack outcome data for our patient cohort, so we cannot determine if RT-qPCR detected PD-L1 is a good indicator for response to anti-PD-1/PD-LI therapy yet. Comparing both methods in patients who are treated with immune check-point inhibitors would reveal more translational conclusions.

## Conclusions

PD-L1 expression in lung cancer has been reported as biomarker that predicts a response to PD-1 inhibitors. However, identification of the major driver mutations in lung cancer patients such as KRAS and EGFR mutations along with expression of PD-L1 would greatly help designing combination treatments for better response. As lung cancer patients harbouring *EGFR* mutations would benefit from EGFR inhibition and the expression of PD-L1 allows for treatment with immune checkpoints blockades such as PD-1/PD-L1 inhibitors. However, only some lung tumors have *EGFR* mutations and PD-L1 levels varies widely. This study found a significant correlation between the absence of *EGFR* mutations and increased PD-L1 expression in patient tumors. This suggests that at least some patients not treatable by EGFR inhibition will benefit from anti PD-1/PD-L1 treatment. Furthermore, our study further found that RT-qPCR has potential as an alternative diagnostic tool to assess the status of PD-L1 expression in the tumors of lung cancer patients.

## Supporting information

**S1 Table. List of primers used in the study.**
(DOCX)

## Author Contributions

**Conceptualization:** Drew Bethune, Mathieu Castonguay, Arik Drucker, Marika Forsythe, Daniel French, Wenda Greer, Harry Henteleff, Mary MacNeil, Paola Marignani, Wojciech Morzycki, Madelaine Plourde, Stephanie Snow, Paola Marcato, Zhaolin Xu.

**Data curation:** Akram Alwithenani, John Fris, Wenda Greer.

**Formal analysis:** Akram Alwithenani, Gordon Flowerdew, John Fris.

**Funding acquisition:** Zhaolin Xu.

**Investigation:** Mathieu Castonguay, Wenda Greer, Paola Marcato, Zhaolin Xu.

**Methodology:** Akram Alwithenani, Paola Marcato.

**Project administration:** John Fris.

**Supervision:** Paola Marcato, Zhaolin Xu.

**Writing – original draft:** Akram Alwithenani.

**Writing – review & editing:** Drew Bethune, Mathieu Castonguay, Arik Drucker, Gordon Flowerdew, Marika Forsythe, Daniel French, Wenda Greer, Harry Henteleff, Mary MacNeil, Paola Marignani, Wojciech Morzycki, Madelaine Plourde, Stephanie Snow, Paola Marcato, Zhaolin Xu.

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
