## [Decision Letter · Decision Letter 0]

8 Feb 2021

PONE-D-20-33968

Profiling targeted driver mutations with PD-L1 expression in non-small cell lung cancer reveals associations with EGFR mutations and vascular invasion

PLOS ONE

Dear Dr. Xu,

Thank you for submitting your manuscript to PLOS ONE. After careful consideration, we feel that it has merit but does not fully meet PLOS ONE’s publication criteria as it currently stands. Therefore, we invite you to submit a revised version of the manuscript that addresses the points raised during the review process.

We look forward to receiving your revised manuscript.

Kind regards,

Srikumar Chellappan

Academic Editor

PLOS ONE

Additional Editor Comments (if provided):

The reviewers have provided their comments. While they are enthusiastic about the study, they has suggested modifications to the manuscript. Please address all the issues; majority of the comments are concerning clarifications and the presentation of the study. We look forward to receiving the revised manuscript.

Journal Requirements:

2)  Thank you for stating the following in the Competing Interests section:

[The authors have declared that no competing interests exist.].

We note that you received funding from a commercial source: Pfizer Canada, Roche Canada, Boehringer

Ingelheim Canada and Merck Canada.

3) In the ethics statement in the manuscript and in the online submission form, please provide additional information about the patient records/samples used in your retrospective study, including: a) whether all data were fully anonymized before you accessed them; b) the date range (month and year) during which patients' medical records/samples were accessed.

4) Please provide the source, product number and any lot numbers of the antibodies purchased for your study.”

5) Please provide the sequences of the primers used in your PCR experiments.

Reviewers' comments:

Reviewer's Responses to Questions

**Comments to the Author**

1. Is the manuscript technically sound, and do the data support the conclusions?

Reviewer #1: Yes

Reviewer #2: Partly

2. Has the statistical analysis been performed appropriately and rigorously? 

Reviewer #1: Yes

Reviewer #2: I Don't Know

3. Have the authors made all data underlying the findings in their manuscript fully available?

Reviewer #1: Yes

Reviewer #2: Yes

4. Is the manuscript presented in an intelligible fashion and written in standard English?

Reviewer #1: Yes

Reviewer #2: Yes

5. Review Comments to the Author

Reviewer #1: The authors aimed to compare the PCR testing and immunohistochemistry (IHC) of PD-L1 in the present study and showed a significant association between them. In addition, they also presented the association among driver gene mutation, tumor pathological feature, and PD-L1 expression. I suggest minor revisions in this paper.

Comments

Introduction

Authors hypothesized the association between EGFR mutation and pathological features based on the fact that those with EGFR mutant NSCLC show a longer survival than EGFR wild type NSCLC. However, longer survival in EGFR mutant NSCLC is contributed by treatment with EGFR-TKI rather than EGFR gene mutation itself. Thus, I would like authors to reconsider whether the above logic is appropriate, although it has been shown that EGFR mutant NSCLC was less vascular invasion. Furthermore, the previous study of reference number 12 investigated the association between post progression survival and overall survival in patients with EGFR mutant NSCLC, did not compare the survival between patients with EGFR wild type and mutant NSCLC. So, it seems to be inappropriate to cite this paper in this context.

Methods

Please describe who evaluated IHC findings.

Please describe the reason for using two statistical software.

Results

The correlation coefficient between the result of PD-L1 PCR testing and IHC is relatively low. Is it possible to create ROC curve and calculate the sensitivity and specificity of a PCR testing for IHC?

I can’t understand how the “expected” was calculated in table 2. Can authors simply show the number and percentage of patients in each category?

Discussion

Authors demonstrated that EGFR mutations are associated with the absence of PD-L1 in the present study. Because many previous authors have reported the association between them, I would like authors to discuss citing meta-analysis by Zhang M et al. (SCienTifiC Reports | 7: 10255 | DOI: 10.1038 / s41598-017-10925-7) and Li D et al. (Eur J Surg Oncol. 2017 Jul; 43 (7): 1372-1379. Doi: 10.1016 / j.ejso. 2017.02.008.)

Reviewer #2: Alwithenani et al. present their manuscript "Profiling targeted driver mutations with PD-L1 expression in non-small cell lung cancer reveals associations with EGFR mutations and vascular invasion" describing the relationship between mutation status and clinical characteristics, PLD1 expression and clinical characteristics, and PDL1 IHC vs qPCR. The strength of the study is the large cohort from which the authors are able to draw a number of conclusions. I believe the title does not accurately reflect the data as PDL1 expression was found to correlate with no EGFR mutation, further as there was no extensive analysis done on vascular invasion, I do not feel this is appropriate for the title. The title should reflect the relationship between mutation status, clinical characteristics IHC, and qPCR. Overall the manuscript is of interest, however, the study is very similar to a study by Tsimateyeu et al (Science Reports 2020) which should be referenced here in. In addition, additional detail should be provided for the methodology used. Major and minor comments are listed below:

Major comments

1. Please explain why signal intensity was not included for the IHC staining analysis.

2. Please reference previous reports using a 1% IHC cutoff

3. There seems to be some discrepancy between the statistical analyses described in the methods section and the methods described below the tables (ie Agretzi z-test table 2, Pearsons Table 3).

4. Please expand on "significant associations between mutations and lymph-vascular invasion and tumor grade could indicate a poor or good prognostic status" this should be able to be determined based on the data

5. The results do not always appropriately reflect the type of test used, please be very specific if you are comparing 2 variables (ie mutation yes or no) or multiple (ie Stage 1-4).

6. It would be helpful to see analysis of IHC positivity as a continuous variable when compared to clinical characteristics allowing it.

7. Why are all discussed driver mutations not included in Table 2?

8. The data in Figure 1 is interesting, although the authors state statistical significance for the EGFR mutations and not the Kras mutations, the differences on the graphs appear the opposite. Can the author further clarify how the analyses for these data were performed`.

9. How were "immune cells" selected, were they sorted? Also, the authors mention a 50% cutoff but it is not clear what this references. It is ahrd to understand how they examined PDL1 mRNA in immune cells if there was not sorting/selecting.

Minor Comments

1. The manuscript should be checked throughout for clarity and lack of repetition (ie. in the intro it reads " Tumor cells with high levels of PLD1 have noticeably higher PDL1 expression in comparison to adjacent lung parenchyma"

2. It isn't clear if the mutation sequencing was done for diagnostic purposes or by the researchers, similarly, why were there 2 panels used for EGFR

3. In pt characteristics in the methods, please specify precisely which mutations have been studies instead of "these driver mutations" were all driver mutations considered or just select ones.

4. Should be clear describing the results in the abstract (ie, increase PDL1 positivity was assoc wt EGFR instead of what is written).

5. For table 2, it would be helpful to describe in the results where the "expected"data came from

6. Please be consistent with the table headers (ie "N" vs "n" vs "no")

7. Please explain all variable in Table 6 in the txt as opposed to "and others"

8. The authors should further discuss how they see these analysis contributing to pt diagnosis or treatment. Is it really more feasible to do PCR than IHC, etc?

6. PLOS authors have the option to publish the peer review history of their article (what does this mean?). If published, this will include your full peer review and any attached files.

Reviewer #1: No

Reviewer #2: No

---

## [Author Response · Author response to Decision Letter 0]

12 Mar 2021

Journal Requirements:

RESPONSE: We did according to PLOS ONE‘s requirement.

2) The Competing Interests section:

RESPONSE: Stated in the cover letter

3) In the ethics statement in the manuscript and in the online submission form, please provide additional information about the patient records/samples used in your retrospective study, including: a) whether all data were fully anonymized before you accessed them; b) the date range (month and year) during which patients' medical records/samples were accessed.

RESPONSE: We did (please see line 91-94)

4) Please provide the source, product number and any lot numbers of the antibodies purchased for your study.”

RESPONSE: Source: Dako PDL1-IHC pharmDX22C3, Product number: SK006 (please see line 116)

5) Please provide the sequences of the primers used in your PCR experiments.

RESPONSE: Please see supplementary table 1 (Table S1)

Reviewer #1 (Reviewer Comments to the Author): 

Introduction

Authors hypothesized the association between EGFR mutation and pathological features based on the fact that those with EGFR mutant NSCLC show a longer survival than EGFR wild type NSCLC. However, longer survival in EGFR mutant NSCLC is contributed by treatment with EGFR-TKI rather than EGFR gene mutation itself. Thus, I would like authors to reconsider whether the above logic is appropriate, although it has been shown that EGFR mutant NSCLC was less vascular invasion. Furthermore, the previous study of reference number 12 investigated the association between post progression survival and overall survival in patients with EGFR mutant NSCLC, did not compare the survival between patients with EGFR wild type and mutant NSCLC. So, it seems to be inappropriate to cite this paper in this context.

RESPONSE: We agree the reviewer’s comment. Indeed, several studies have shown EGFR-TKI had a significant impact on survival. For the clarification we modified the sentence as EGFR mutations are linked with good prognosis in lung cancer patients mainly attributed to the treatment of tyrosine kinase inhibitor. However, those patients are at late stage of the disease, and the patients included in our study are diagnosed at early stage and treated with surgical resection. Izar et al. also showed the significance correlation between EGFR positive patients and survival in surgically resected NSCLC without receiving TKI. This paper is now cited in our manuscript in reference number 17. (please see line 78-79)

Methods

Please describe who evaluated IHC findings.

RESPONSE: All IHC finding were evaluated by lung pathologists (Z.X. and M.C). It has been added to the method section. (Please see line 131-132) 

Please describe the reason for using two statistical software.

RESPONSE: SAS was used because it is the most appropriate software to analyze clinical data for large cohort. Graph prism was used analyze two variable analysis and generate graphs. We have added these clarifications in the manuscript. (Please see line 144-147)

Results

The correlation coefficient between the result of PD-L1 PCR testing and IHC is relatively low. Is it possible to create ROC curve and calculate the sensitivity and specificity of a PCR testing for IHC?

RESPONSE: Owing limited number of cases, it is hard to create ROC curve and draw a conclusion out of it. 

I can’t understand how the “expected” was calculated in table 2. Can authors simply show the number and percentage of patients in each category?

RESPONSE: In Table 2, the Pearson’s goodness-of-fit test analysis were performed. The Pearson’s goodness-of-fit test is a single number that tells you how much difference exists between your observed counts and the counts you would expect if there were no relationship at all in the population. In any given table, expected values are calculated by multiplying the total number of the raw with the total number of the column divided by overall total number of the table. This information has now been added under Table 2. (Please see line 448-450)

Discussion

Authors demonstrated that EGFR mutations are associated with the absence of PD-L1 in the present study. Because many previous authors have reported the association between them, I would like authors to discuss citing meta-analysis by Zhang M et al. (SCienTifiC Reports | 7: 10255 | DOI: 10.1038 / s41598-017-10925-7) and Li D et al. (Eur J Surg Oncol. 2017 Jul; 43 (7): 1372-1379. Doi: 10.1016 / j.ejso. 2017.02.008.)

RESPONSE: As requested, we have discussed these two papers. (Please see lines 298-302)

Reviewer #2 (Reviewer Comments to the Author): 

Alwithenani et al. present their manuscript "Profiling targeted driver mutations with PD-L1 expression in non-small cell lung cancer reveals associations with EGFR mutations and vascular invasion" describing the relationship between mutation status and clinical characteristics, PLD1 expression and clinical characteristics, and PDL1 IHC vs qPCR. The strength of the study is the large cohort from which the authors are able to draw a number of conclusions. I believe the title does not accurately reflect the data as PDL1 expression was found to correlate with no EGFR mutation, further as there was no extensive analysis done on vascular invasion, I do not feel this is appropriate for the title. The title should reflect the relationship between mutation status, clinical characteristics IHC, and qPCR. Overall the manuscript is of interest, however, the study is very similar to a study by Tsimateyeu et al (Science Reports 2020) which should be referenced here in. In addition, additional detail should be provided for the methodology used. Major and minor comments are listed below:

RESPONSE: The title has been changed to “Profiling non-small cell lung cancer reveals that PD-L1 is associated with wild type EGFR and vascular invasion, and immunohistochemistry quantification of PD-L1 correlates weakly with RT-qPCR”. (please see lines 1-3) The paper by Tsimateyeu et al (Science Reports 2020) has been added in the reference number 36.

Major comments

1. Please explain why signal intensity was not included for the IHC staining analysis.

RESPONSE: PD-L1 staining was performed using the FDA and Health Canada approved pharmDx22C3 kit. Thus, PD-L1 protein expression is determined by a Tumor Proportional Score (TPS), which is the percentage of viable tumor cells showing partial or complete membrane staining. The intensity of the stain does not affect TPS scoring. 

2. Please reference previous reports using a 1% IHC cutoff.

RESPONSE: We used 1% and 50% cut-offs for PD-L1 expression to align with current clinical practice and clinical significance. (please see lines 130-131). Two references have been added in number 39 and 40.

3. There seems to be some discrepancy between the statistical analyses described in the methods section and the methods described below the tables (ie Agretzi z-test table 2, Pearsons Table 3).

RESPONSE: Originally, we used Agretzi z-test to analyze all variables, but we realized that the Pearson’s goodness-of-fit test and Fisher exact test could simply provide us with answers needed. Now, we have made all changes needed (please see lines 144-147). 

4. Please expand on "significant associations between mutations and lymph-vascular invasion and tumor grade could indicate a poor or good prognostic status" this should be able to be determined based on the data.

RESPONSE: We have clarified this point in the text (please see lines 182-186)

5. The results do not always appropriately reflect the type of test used, please be very specific if you are comparing 2 variables (ie mutation yes or no) or multiple (ie Stage 1-4).

RESPONSE: Good point. We have added more description under Tables. 

6. It would be helpful to see analysis of IHC positivity as a continuous variable when compared to clinical characteristics allowing it.

RESPONSE: PD-L1 IHC assessment by TPS is a semiquantitative method using the FDA and Health Canada approved pharmDx22C3 kit for PD-L1 staining. Therefore, we followed the current clinical practice to choose the cut-offs instead of assessing a continuous variable. 

7. Why are all discussed driver mutations not included in Table 2?

RESPONSE: Table 2 is a summary of all significant associations between variables and gene mutations in the lung cancer patient cohort. However, other discussed driver mutations are analyzed in Tables 3, 4 and 5. We have clarified this in the Results section (please see lines 177-178).

8. The data in Figure 1 is interesting, although the authors state statistical significance for the EGFR mutations and not the Kras mutations, the differences on the graphs appear the opposite. Can the author further clarify how the analyses for these data were performed`.

RESPONSE: In Figure 1, a fisher exact test was used to determine if there are non-random associations between two categorical variables (PD-L1 and EGFR or KRAS). In Figure 1A, EGFR positive patients were shown to negatively correlated to PD-L1 (26% of EGFR+ patients had PD-L1 expression versus 74% had PD-L1 expression in the same population. We have clarified this in the legend of Figure 1 (please see lines 521-523).

9. How were "immune cells" selected, were they sorted? Also, the authors mention a 50% cutoff but it is not clear what this references. It is hard to understand how they examined PDL1 mRNA in immune cells if there was not sorting/selecting.

RESPONSE: In the RT-qPCR analysis, RNA was extracted from the unsorted tumor samples that include immune cells within the specimens. Technically, it is impossible to separate tumor cells and immune cells in a RT-qPCR analysis. Although this is a suboptimal situation, we found correlations between mRNA and IHC protein expression of PD-L1, presumably the amount of admixed immune cells within the specimens not significantly affecting overall PD-L1 mRNA assessment in such a setting. We currently lack outcome data for our patient cohort, so we cannot determine if RT-qPCR detected PD-L1 is a good indicator for response to PD1/PD-L1 therapy. Comparing both methods in patients who are treated with immune check-point inhibitors would reveal more translational conclusions. This information has now been added to the manuscript. (Please see line 323-329). The cut-offs (such as 1% and 50%) for the PD-L1 IHC evaluation in this study are aligned with current clinical practice using pharmDx22C3.

Minor Comments

1. The manuscript should be checked throughout for clarity and lack of repetition (ie. in the intro it reads "Tumor cells with high levels of PLD1 have noticeably higher PDL1 expression in comparison to adjacent lung parenchyma"

RESPONSE: We made changes. (Please see line 73-74) 

2. It isn't clear if the mutation sequencing was done for diagnostic purposes or by the researchers, similarly, why were there 2 panels used for EGFR

RESPONSE: The mutation analysis was performed for both diagnostic and research purposes. For EGFR, two panels were used as we needed to analyze point mutation (such as L858R, mentioned in Table 1) and also detect deletion and insertion (such as Exon 19 deletions and Exon 20 insertions; mentioned in table 1) in the gene. 

3. In pt characteristics in the methods, please specify precisely which mutations have been studies instead of "these driver mutations" were all driver mutations considered or just select ones.

RESPONSE: We have clarified this point in the text. (please see lines 158-159)

4. Should be clear describing the results in the abstract (ie, increase PDL1 positivity was assoc wt EGFR instead of what is written).

RESPONSE: We have modified this sentence in the abstract to be: Specifically, increased PD-L1 expression was associated with wildtype EGFR and vascular invasion, and total PD-L1 mRNA levels correlated weakly with protein expression on tumor cells. (Please see line 33-35)

5. For table 2, it would be helpful to describe in the results where the "expected"data came from

RESPONSE: We have provided clarification under Table 2. In any given table where the Pearson’s goodness-of-fit test is applied, the expected values are calculated by multiplying the total number of the raw with the total number of the column divided by overall total number of the table.

6. Please be consistent with the table headers (ie "N" vs "n" vs "no")

RESPONSE: We have made all changes in the table headers. 

7. Please explain all variable in Table 6 in the txt as opposed to "and others"

RESPONSE: We have explained the other variables in the Table and in the text. (Please see line 205-209)

8. The authors should further discuss how they see these analysis contributing to pt diagnosis or treatment. Is it really more feasible to do PCR than IHC, etc?

RESPONSE: We have further discussed the contribution of the analysis and the feasibility of qPCR as diagnostic tools. (Please see lines 323-329)

---

## [Decision Letter · Decision Letter 1]

30 Mar 2021

PONE-D-20-33968R1

Profiling non-small cell lung cancer reveals that PD-L1 is associated with wild type EGFR and vascular invasion, and immunohistochemistry quantification of PD-L1 correlates weakly with RT-qPCR

PLOS ONE

Dear Dr. Xu,

Thank you for submitting your manuscript to PLOS ONE. After careful consideration, we feel that it has merit but does not fully meet PLOS ONE’s publication criteria as it currently stands. Therefore, we invite you to submit a revised version of the manuscript that addresses the points raised during the review process.

We look forward to receiving your revised manuscript.

Kind regards,

Srikumar Chellappan

Academic Editor

PLOS ONE

Journal Requirements:

Additional Editor Comments (if provided):

The authors have addressed all the issues raised in the original review. One minor change has been suggested by one reviewer; please make this change and highlight it. A decision can be made at the editorial level once this change is made.

Reviewers' comments:

Reviewer's Responses to Questions

**Comments to the Author**

1. If the authors have adequately addressed your comments raised in a previous round of review and you feel that this manuscript is now acceptable for publication, you may indicate that here to bypass the “Comments to the Author” section, enter your conflict of interest statement in the “Confidential to Editor” section, and submit your "Accept" recommendation.

Reviewer #1: (No Response)

Reviewer #2: All comments have been addressed

2. Is the manuscript technically sound, and do the data support the conclusions?

Reviewer #1: Yes

Reviewer #2: (No Response)

3. Has the statistical analysis been performed appropriately and rigorously? 

Reviewer #1: Yes

Reviewer #2: (No Response)

4. Have the authors made all data underlying the findings in their manuscript fully available?

Reviewer #1: Yes

Reviewer #2: (No Response)

5. Is the manuscript presented in an intelligible fashion and written in standard English?

Reviewer #1: Yes

Reviewer #2: (No Response)

6. Review Comments to the Author

Reviewer #1: Authors have appropriately replied to review comments. However, in discussion, authors described as below; "Indeed, two meta-analysis studies have reported lack of associations between PD-L1 presence and EGFR mutations in lung cancer patients [33, 34]." The reference number 34 reported that there were the association between PD-L1 expression and EGFR mutation status, so the above description is thought to be inappropriate.

Reviewer #2: (No Response)

7. PLOS authors have the option to publish the peer review history of their article (what does this mean?). If published, this will include your full peer review and any attached files.

Reviewer #1: No

Reviewer #2: No

---

## [Author Response · Author response to Decision Letter 1]

31 Mar 2021

6. Review Comments to the Author

Reviewer #1: Authors have appropriately replied to review comments. However, in discussion, authors described as below; "Indeed, two meta-analysis studies have reported lack of associations between PD-L1 presence and EGFR mutations in lung cancer patients [33, 34]." The reference number 34 reported that there were the association between PD-L1 expression and EGFR mutation status, so the above description is thought to be inappropriate.

We modified the sentence as “Some studies have reported lack of associations between PD-L1 expression and EGFR status in lung cancer patients [33]” and removed reference [34] there. It is followed by “In this study we found that ….”. These two sentences are switched the order (comparing to the previous version) to make a more clear flow.

---

## [Editor Report · Decision Letter 2]

20 Apr 2021

Profiling non-small cell lung cancer reveals that PD-L1 is associated with wild type EGFR and vascular invasion, and immunohistochemistry quantification of PD-L1 correlates weakly with RT-qPCR

PONE-D-20-33968R2

Dear Dr. Xu,

We’re pleased to inform you that your manuscript has been judged scientifically suitable for publication and will be formally accepted for publication once it meets all outstanding technical requirements.

Kind regards,

Srikumar Chellappan

Academic Editor

PLOS ONE
---

## [Editor Report · Acceptance letter]

27 Apr 2021

PONE-D-20-33968R2 

Profiling non-small cell lung cancer reveals that PD-L1 is associated with wild type EGFR and vascular invasion, and immunohistochemistry quantification of PD-L1 correlates weakly with RT-qPCR 

Dear Dr. Xu:

I'm pleased to inform you that your manuscript has been deemed suitable for publication in PLOS ONE. Congratulations! Your manuscript is now with our production department. 

Kind regards, 

on behalf of

Dr. Srikumar Chellappan 

Academic Editor

PLOS ONE